# The Effect of Hyaluronic Acid and Chondroitin Sulphate-Based Medical Device Combined with Acid Suppression in the Treatment of Atypical Symptoms in Gastroesophageal Reflux Disease

**DOI:** 10.3390/jcm11071890

**Published:** 2022-03-29

**Authors:** Gaia Pellegatta, Benedetto Mangiavillano, Rossella Semeraro, Francesco Auriemma, Elisa Carlani, Alessandro Fugazza, Edoardo Vespa, Alessandro Repici

**Affiliations:** 1Endoscopy Unit, Department of Gastroenterology, IRCCS Humanitas Research Hospital, Via Manzoni 56, Rozzano, 20089 Milan, Italy; rossella.semeraro@gmail.com (R.S.); elisa.carlani@humanitas.it (E.C.); alessandro.fugazza@humanitas.it (A.F.); edoardo.vespa@humanitas.it (E.V.); alessandro.repici@hunimed.eu (A.R.); 2Gastrointestinal Endoscopy Unit, Humanitas Mater Domini, Via Gerenzano 2, 21053 Castellanza, Italy; benedetto.mangiavillano@mc.humanitas.it (B.M.); francesco.auriemma@materdomini.it (F.A.); 3Department of Biomedical Sciences, Humanitas University, Via Rita Levi Montalcini 4, Pieve Emanuele, 20090 Milan, Italy

**Keywords:** gastroesophageal reflux disease, extraesophageal reflux symptoms, hyaluronic acid, chondroitin sulphate

## Abstract

Extraesophageal reflux symptoms are increasingly common in the Western population and their clinical management is still controversial. Although therapy with proton-pump inhibitors (PPIs) represents the gold standard, to date, many patients are refractory to this treatment. The aim of this study was to evaluate, in patients with a recent diagnosis of GERD experiencing extraesophageal symptoms, the efficacy and safety of a 6-week treatment with PPI acid suppression in combination with Gerdoff^®^ (a hyaluronic acid and chondroitin sulphate-based medical device) compared to PPI monotherapy. The trial verified the reduction in symptom frequency and severity by evaluating the proportion of Responders and Non-Responder patients after 6 weeks of treatment, compared to baseline. The effects of Gerdoff^®^ + PPI treatment on extraesophageal symptoms were also evaluated after a 12-week follow up only in Responder patients. The analysis of the change in total Reflux Symptoms Index (RSI) score from baseline to the other time points showed that the extent of the decrease from baseline was higher in the Gerdoff^®^ + PPI group than in the PPI group at any time point. However, the comparison between groups did not show statistically significant differences at any time point. A statistically significant difference, in favor of the Gerdoff^®^ + PPI group, was observed for individual RSI items. Even if the trial showed some limitations, this is the first published study on the efficacy of a medical device containing hyaluronic acid and chondroitin sulphate with antacid in the treatment of extraesophageal reflux symptoms.

## 1. Introduction

Gastroesophageal reflux disease (GERD) is associated with the reflux of gastric acid or intestinal bile contents into the esophagus. 

According to the Montreal Consensus Conference, typical GERD symptoms are heartburn and regurgitation [1]. The extraesophageal symptoms that have been ascribed to GERD include pulmonary diseases (e.g., asthma, bronchitis, microaspiration, and pulmonary fibrosis), ear, nose, and throat (ENT) symptoms (e.g., hoarseness, cough, laryngitis, subglottic stenosis, and laryngeal cancer), as well as non-cardiac chest pain, dental erosions, sinusitis, pharyngitis, and sleep apnea [1]. GERD may contribute to extraesophageal syndromes through either a direct or an indirect (vagal-mediated) mechanism. More specifically, injury can result from the direct effect of gastric juice on the mucosa of the tracheobronchial tree, the laryngopharynx, including the vocal cords, the middle ear, and the nasal sinus complex, or from the macro- and microaspiration of refluxed gastroduodenal contents [2].

An association between gastroesophageal reflux and laryngeal symptoms is supported by frequent observations of these symptoms in patients with GERD. An Italian study demonstrated that 74.4% of patients with GERD present at least one extra-esophageal symptom and laryngeal symptoms are present in a high proportion of patients (19.9–38.7%) [3]. The causes of pulmonary, pharyngeal, and laryngeal symptoms, other than GERD, are voice abuse, smoking, alcohol, infectious disease, and allergy. Moreover, asymptomatic gastro-esophageal reflux was demonstrated in between 50% and 75% of subjects with chronic cough [4]. The link between reflux and cough has been confirmed by the disappearance of episodes of nocturnal cough following appropriate therapy for gastroesophageal reflux.

It was estimated that between 4% and 10% of patients attending ear, nose, and throat (ENT) specialist consultations report signs and symptoms associated with GERD [1]. It should be pointed out that, in recent years, visits performed by ENT specialists due to upper GI symptoms have increased by 500% [5] and that upper reflux is present in more than 50% of patients with dysphonia [6].

Acid suppression therapy with proton pomp inhibitors (PPI) is the current standard of care for patients with GERD with typical and atypical presentations. However, the effectiveness of PPI therapy in patients with extraesophageal symptoms is less robust than in subjects with typical GERD manifestations. It has been demonstrated that 50% of patients with atypical GERD symptoms do not respond to 8–12 weeks of PPI therapy and 15% show only partial response [7,8].

Moreover, it is widely accepted that the mucosal lesions caused by GERD have to be treated with PPI therapy combined with other active substances or devices able to potentiate the effects of the PPI and, thus, improve mucosal defenses [8,9]. These devices improve mucosal defenses by creating a film over the esophageal mucosa and acting as a mechanical barrier against the noxious components of both acidic and basic refluxate [10]. Although the use of these new medical devices in addition to PPI therapy is expected to improve the treatment of patients with extraesophageal GERD symptoms, the literature is still lacking data to prove it.

Emerging data are emphasizing the role of new medical devices containing hyaluronic acid (HA) and chondroitin sulphate (CS) with an antacid component in the treatment of GERD. Gerdoff^®^ is a class-3 CE-marked medical device containing CS, HA, and aluminum hydroxide (HA:AH:CS present with a ratio of 1:20:40) with a melt-in-mouth tablet formulation.

The aim of this study was to evaluate, in patients with a recent diagnosis of GERD experiencing extraesophageal symptoms, the efficacy and safety of a 6-week treatment with PPI acid suppression in combination with Gerdoff^®^ compared to PPI monotherapy. Furthermore, the long-term therapeutic efficacy of Gerdoff^®^ on extraesophageal symptoms was evaluated over a 12-week follow-up period. Patients included in the follow-up period (Gerdoff^®^ + PPI responders only) were randomized to receive Gerdoff^®^ or no treatment.

## 2. Materials and Methods

### 2.1. Patients

Patients with extraesophageal symptoms and a recent diagnosis of GERD were enrolled in the study. The diagnosis of GERD with extraesophageal symptoms was based on clinical presentation (hoarseness, cough, throat clearing, sore throat, voice changes, globus sensation, and postnasal drip) and positivity on a validated questionnaire, the Reflux Symptoms Index (RSI), with a total score of >20 [11]. The RSI is a self-administrated questionnaire and examines nine items, to be scored from 0 to 5, in which a higher score indicates greater symptom severity.

Frequency of extraesophageal symptoms was evaluated using a self-administrated Likert scale questionnaire. A satisfaction judgment on the treatment’s effect on extraesophageal symptoms was also obtained using a 4-item scale (1-poor, 2-fair, 3-good, 4-excellent).

Extraesophageal GERD symptoms had to have been present for at least 3 months and could be associated or not with typical GERD symptoms (e.g., regurgitation and heartburn). Patients had to have been free from anti-secretory medication (either PPI or histamine 2 receptor antagonists (H2RA)), antacids, and alginate-containing formulations for at least 4 weeks prior to enrolment in the trial. The patients enrolled were of both sexes and aged between 18 and 75.

Exclusion criteria were presence of infective or chemical esophagitis, acute or chronic nasosinusitis, chronic bronchitis, and diagnosis of gastrointestinal cancer. Patients on chronic therapy with drugs affecting salivary secretion (e.g., antihistamines or inhaled corticosteroids) were not enrolled.

### 2.2. Study Design

This was a multicenter, prospective, randomized, open-label, two-parallel-group study, followed by a follow-up period.

Two Italian hospitals were involved in the study and each of them obtained the approval of the competent ethics committee. The study was registered at ClinicalTrials.gov as NCT03793556. The trial was performed in accordance with Good Clinical Practice (GCP) guidelines [12] and the Declaration of Helsinki [13].

Eligible patients gave informed and written consent and were randomized (1:1) to receive open-label treatment with Gerdoff^®^ three times a day plus Omeprazole 40 mg (PPI) once a day (Group 1) or Omeprazole 40 mg (PPI) once a day alone (Group 2) for 6 weeks.

During the treatment period, no rescue therapy with anti-secretory medications (either PPI or H2RA), antacids, or alginate-containing formulations was permitted, and any use of these drugs had to be registered.

At the end of this short-course therapy, patients enrolled in the Gerdoff^®^ + PPI Group were classified as Responders and Non-Responders on the basis of the RSI questionnaire (responders = at least a 50% reduction in RSI score compared to baseline). To assess the maintenance of the therapeutic effect, responders were randomized 1:1 to open-label treatment with Gerdoff^®^ or to the control group, which did not receive any treatment for 12 weeks. During this study phase, rescue therapy with Omeprazole according to a standard and constant dose regimen was permitted.

Safety and tolerability were assessed during each visit by recording all adverse events, defined as any unfavorable or unintended symptom and/or sign, considered to have a causal relationship with the drugs used in the study. The study design and the detailed assessment schedule are shown in Figure 1.

### 2.3. Statistical Analysis

The primary endpoint of the study was treatment efficacy, calculated as the variation in the total RSI score between the baseline and week 6 in the two groups. In the analysis of the primary performance variable, the two treatment groups were compared using Student’s *t*-test for independent data. Sensitivity analyses were conducted on the primary performance variable. The study of the temporal profile of the RSI questionnaire was performed using repeated measures analysis of variance (ANOVA). In addition to the treatment effect, the time effect and the treatment by time interaction were included in the model.

The secondary endpoints of the study were: (I) change in total RSI score between baseline and the other intermediate time points; (II) change in individual RSI item scores between baseline and any time point; (III) number and percentage of responder/non-responder patients at end of treatment (visit V4); (IV) change in upper symptoms, assessed using RSI score and the Likert scale, between baseline and end of treatment (visit V4) and end of follow-up (visit V6); (V) use of rescue medication during the follow-up period: administered therapy, frequency of administration, timing of administration, and administered dose; (VI) patient satisfaction with treatment, rated at intermediate visits, and at the end of treatment visit, by means of a semiquantitative ordinal scale, where: 0 = poor, 1 = fair, 2 = good, 3 = excellent.

Yates’ chi-squared test was used to compare the distribution of the proportions of responders and non-responders in the two treatment groups.

In the first 6 weeks of treatment, the intake of permitted or prohibited rescue medication in each treatment group was calculated as the frequency of administration of the specific drug and the comparison between groups was performed using Fisher’s exact test. In the first 6 weeks of treatment, the median dose of rescue medication in the two study groups was compared using the non-parametric (distribution-free) median test. If a mean intake value could also be calculated, the comparison between the two treatment groups was performed using the Mann–Whitney U test. The judgment frequencies were compared between the groups using the Mantel–Haenszel chi-squared test.

A sample size of 62 patients (78 patients after 20% drop-out rate correction) would provide 80% power to demonstrate the superiority of Gerdoff^®^ + PPI versus PPI alone assuming a 3.6-point change in RSI score from baseline between treatments with 5 points in SD for two-sided unpaired Student’s *t*-test after 6 weeks of treatment.

All *p*-values reported were two-sided. Results were considered to be statistically significant if the two-sided *p*-value was <0.05.

## 3. Results

### 3.1. Patient Baseline Characteristics

A total of 71 patients were included and randomized to the assigned treatment group: 35 patients received Gerdoff^®^ + omeprazole and 36 omeprazole monotherapy (Table 1). Thirteen (37.1%) patients in the Gerdoff^®^ + omeprazole group and 2 (5.6%) in the omeprazole group dropped out of the study during the 6-week treatment period due to withdrawal of consent. There were no dropouts due to adverse events. In the follow-up period, 18 patients were randomized to the two groups: 10 patients received Gerdoff^®^ and 8 received no treatment. Two patients who were randomized to treatment with Gerdoff^®^ did not attend all the visits scheduled for the follow-up period (Table 1). There were no important differences between groups in terms of demographic and baseline characteristics and vital signs, except for the distribution of gender (Table 1). The Gerdoff^®^ + omeprazole group included a significantly higher proportion of males (37.1%) than the omeprazole group (11.1%) (Table 1). Seven patients took non-permitted rescue medication (six in the omeprazole group and one in the Gerdoff^®^ + omeprazole group) and were excluded from the sensitivity analysis on the primary endpoint (Table 1). All patients in both groups showed >95% compliance during the 6-week treatment period and follow-up period.

### 3.2. Total RSI Score during Treatment Period

With regard to the primary endpoint, Table 2 shows that the mean and median values of the total RSI score progressively decreased from baseline to the end of the 6-week treatment period in both groups.

The adjusted mean change in the total RSI score between baseline and week 6 was −16.2 (7.45) in the Gerdoff^®^ + omeprazole group and −13.7 (10.58) in the omeprazole group. Although the adjusted mean change in total RSI score between baseline and week 6 was slightly higher in the Gerdoff^®^ + omeprazole group than in the omeprazole group, this difference was not statistically significant (*p* = 0.2760 in the unpaired *t*-test, *p* = 0.2679 with Cochran correction—Table 3).

During the follow-up period after the 6-week treatment, the mean total RSI score decreased slightly between the start and the end of the follow-up phase in the Gerdoff^®^ group and did not change substantially in the untreated group (Table 4).

### 3.3. Change in Total RSI Score from Baseline to the Other Timepoints

The analysis of the change in total RSI score from baseline to the other time points (V2; week 1; V3, week 3) showed that the decrease from baseline to any time point was greater in the Gerdoff^®^ + omeprazole group than in the omeprazole group (Table 5).

A further analysis, performed using an ANOVA model to examine the entire curve profile, showed a statistically significant treatment effect (F value = 6.13, *p* = 0.0157) and a statistically significant visit effect (F value = 73.00, *p* < 0.0001), whereas the treatment-by-visit interaction was not statistically significant (F value = 1.07, *p* = 0.3695). The analysis performed by stratifying the treatment effect by visit resulted in statistically significant treatment effects at week 1 (V2) (F value = 5.03, *p* = 0.0281) and week 6 (V4) (F value = 6.39, *p* = 0.0138), whereas no statistically significant treatment effects were observed at baseline (V1) (F value = 2.16, *p* = 0.1459) and week 3 (V3) (F value = 1.87, *p* = 0.1754). In the analysis performed by stratifying the treatment effect according to overall visits, a statistically significant effect was observed in both treatment groups (*p* < 0.0001 in both groups).

### 3.4. Change in the Individual RSI Item Score from Baseline to Any Time Point

As a secondary endpoint, the trend for the individual RSI item scores showed a progressive decrease in mean and median scores for all items from baseline to the end of the 6-week treatment period in both groups (Table 6). The comparison between groups showed a statistically significant difference, in favor of the Gerdoff^®^ + omeprazole group, for the items ‘cough after eating or after lying down’ (*p* = 0.0027 between groups), ‘troublesome or annoying cough’ (*p* = 0.0443 between groups), and ‘heart burn, chest pain, indigestion, or stomach acid in the mouth’ (*p* = 0.0240 between groups) at week 6 (V4). Other investigated items (e.g., difficulty swallowing food, liquids, or pills; breathing difficulties or choking episodes) were not found to be significantly different (Appendix A).

### 3.5. Number and Percentage of Responder/Non-Responder Patients at Week 6

The Week-6 responder rate was significantly higher in the Gerdoff^®^ + omeprazole group than in the omeprazole group. The number and percentage of responders at Week 6 were 25 (81%) in the Gerdoff^®^ + omeprazole group and 21 (58%) in the omeprazole group and the difference between groups was statistically significant (*p* = 0.0496 in the Chi-squared test) (Figure 2).

### 3.6. Change in the Frequency of Extraesophageal Symptoms Assessed Using the Likert Scale from Baseline to End of Treatment (Week 6) and End of Follow-Up (Week 18)

The results of the extraesophageal GERD symptom assessments using the Likert scale were consistent with those reported for the RSI questionnaire. More specifically, the mean and median values or total Likert scale score progressively decreased from baseline to the end of the 6-week treatment period in both groups without evidence of significant differences between the groups (*p* = 0.0547 Student’s *t*-test with Cochran correction) (Table 7). The results of the correlation test between the total RSI score and the total Likert scale score showed a high and statistically significant level of correlation (Pearson’s r coefficient = 0.921, *p* < 0.0001).

### 3.7. Use of Rescue Medication during the Follow-Up Period: Administered Therapy, Frequency of Administration, Timing of Administration, and Administered Dose

More patients used omeprazole as rescue medication in the omeprazole group (77.8%) than in the Gerdoff^®^ + omeprazole group (68.6%) during the 6-week treatment period. Seven patients (77.9% of evaluable patients) in the Gerdoff^®^ group and five (62.5% of evaluable patients) in the untreated group used omeprazole as rescue medication during the follow-up period. None of the patients (0.0%) in the Gerdoff^®^ + omeprazole group and four patients (11.4%) in the omeprazole group used other drugs as rescue medication.

### 3.8. Patient-Reported Satisfaction with Treatment

The majority of patients in both groups gave an excellent or a good satisfaction judgment for treatment at any post-baseline timepoint (Appendix A).

### 3.9. Adverse Events (AEs)

Overall, both the Gerdoff^®^ + omeprazole combination and omeprazole monotherapy were very well tolerated. The number of AEs and the percentage of patients who experienced AEs were similar in the two groups. There was no evidence of important changes in any vital parameters from baseline to any post-baseline time-point in either group. A total of 28 AEs were reported in 14 patients (40.0%) in the Gerdoff^®^ + omeprazole group and 29 AEs were reported in 12 patients (33.3%) in the omeprazole group. No fatal adverse events occurred in any patient (Table 8). The most common AEs by preferred term were nausea, with four AEs (14.3% of all AEs) in the Gerdoff^®^ + omeprazole group and three AEs (10.3%) in the omeprazole group, and influenza-like illness, with four AEs (14.3% of all AEs) in the Gerdoff^®^ + omeprazole group and two AEs (5.6%) in the omeprazole group. None of the reported AEs were considered related to treatment with Gerdoff^®^ in either treatment group.

## 4. Discussion

The results of this study showed that in patients with extraesophageal symptoms and a recent diagnosis of GERD, treatment with Gerdoff^®^ + omeprazole leads to a greater, though not significantly so, reduction in symptoms than with PPI therapy alone. The primary endpoint results showed that the adjusted mean change in total RSI score from baseline to visit V4 (week 6) was slightly higher in the Gerdoff^®^ + omeprazole group than in the omeprazole group. However, the hypothesis of the superiority of Gerdoff^®^ + omeprazole over omeprazole monotherapy was not confirmed. The adjusted mean difference between the Gerdoff^®^ + omeprazole group and the omeprazole group was −2.5 (9.27) and the 95% CI of the difference in adjusted means was −7.03 to 2.04, showing that the difference between groups was not statistically significant (*p* = 0.2760 in the unpaired *t*-test, *p* = 0.2679 with Cochran correction). The results of the sensitivity analysis excluding the seven patients who took non-permitted rescue medication (one in the Gerdoff^®^ + omeprazole group and six in the omeprazole group) were consistent with those observed in the primary analysis. The analysis of the change in total RSI score from baseline to the other time points showed that the extent of the decrease from baseline was higher in the Gerdoff^®^ + omeprazole group than in the omeprazole group at any time point. However, the comparison between groups did not show statistically significant differences at any time point. In the interpretation of results, it should be considered that, during the 6-week treatment period, a considerable proportion of patients in both treatment groups used omeprazole or other drugs as rescue medication (this proportion was higher in the omeprazole monotherapy group than in the combination group), which might have reduced the potential difference between groups in the primary and secondary performance endpoints. This result supports the hypothesis that PPI alone is not sufficient for the treatment of extraesophageal GERD symptoms. Although the anticipated number of patients to be enrolled was 78, due to difficulties in identifying eligible patients, recruitment was interrupted at 72. The standard deviation of the difference in adjusted means between the Gerdoff^®^ + omeprazole group and the omeprazole group was 9.27, which was 4.27 points higher than expected. These factors may have contributed to the statistical insignificance of the primary performance endpoint.

The ANOVA test analysis of the RSI scores between the two treatments and visits over time showed statistically significant differences both between the two treatment groups and between visits, although the two time profiles showed a similar trend. Within each individual treatment, there was a significant reduction in the total RSI score. This decrease appeared to be greater in the Gerdoff^®^ + omeprazole group, in which an effect was observed after just one week of treatment (V2), before coming to an equilibrium phase (V3) and a subsequent further decrease (V4). Considering patients with an at least 50% decrease in RSI score from baseline as Responders (V4), a significant difference could be observed between the two groups in favor of treatment with Gerdoff^®^ + omeprazole. The analysis of the total RSI score results in the follow-up period showed that the mean total RSI score decreased slightly between the start and the end of the follow-up phase in the Gerdoff^®^ group and did not change substantially in the untreated group. The effect observed during the treatment period was maintained or further improved during the follow-up phase in both groups, except in one patient in the untreated group, in whom the total RSI scores were 10 and 24 at the start and at the end of the follow-up period, respectively. It is likely that the change in this patient accounted for the apparent maintenance of the mean total RSI score between the start and the end of the follow-up period. The trend in the results for the individual RSI items showed a progressive decrease in the mean and median scores for all items from baseline to the end of the 6-week treatment period in both groups. A statistically significant difference, in favor of the Gerdoff^®^ + omeprazole group, was observed for the items relating to cough and typical GERD symptoms. The results for the upper GERD symptom assessments using the Likert scale were consistent with those reported using the RSI questionnaire. As with the RSI score, the mean and median total Likert scale score values decreased progressively between baseline and the end of the 6-week treatment period in both groups. Similarly, the trend in the results for the individual items of the Likert scale showed a progressive decrease in the mean and median values of all item scores between baseline and the end of the 6-week treatment period in both groups, without any evidence of substantial differences between them. Six patients in the omeprazole group and one in the Gerdoff^®^ + omeprazole group used other rescue medications. The distribution of patient-reported satisfaction with treatment was similar in the two groups. The majority of patients in both groups reported excellent or good satisfaction with treatment at any post-baseline time-point (Appendix A).

The safety results showed that both Gerdoff^®^ and omeprazole were very well tolerated. Few patients in both treatment groups had post-baseline clinically significant blood count or blood chemistry abnormalities and none were treatment related. There was no evidence of considerable changes in any vital parameters from baseline to any post-baseline time-point in either group.

### 4.1. Comparisons with Other Studies

Acid suppression therapy is the first-line treatment for extraesophageal GERD symptoms and represents the general standard of care at the current time. Three months’ therapy with maximal dosage PPI b.i.d. is the first therapeutic approach in clinical practice. If the patient responds to PPI therapy, the minimum effective dose must be maintained in the long-term. It has been demonstrated that, in patients with atypical GERD symptoms, 50% do not respond to 8–12 weeks of PPI therapy and 15% show only partial response [14]. Moreover, PPI treatment in patients with extraesophageal GERD symptoms is based on poor-quality scientific evidence and there are little data to support the superiority of PPIs over placebo. Although many studies have focused on laryngopharyngeal reflux (LPR), laryngitis, asthma, and chronic cough, their results are inconsistent and only a few are randomized controlled trials (RCTs). The few RCTs that have recently been published suggest that PPIs have little or no superiority to placebo in the treatment of extraesophageal GERD symptoms [15,16].

Several reviews and meta-analyses assessing the efficacy of PPIs in patients with suspected LPR have reported mixed results.

One meta-analysis of RCTs demonstrated a significantly greater improvement in LPR symptom alleviation with PPI than with placebo but no differences in treatment response rate (defined as >50% reduction in laryngopharyngeal symptoms) and endoscopic examination findings [14], whereas two earlier meta-analyses of RCTs demonstrated no significant differences in LPR symptom reduction with PPIs compared to placebo [17,18].

A recent systematic review demonstrated that six out of the nine systematic reviews/meta-analyses concluded that PPI therapy is not superior to placebo and three concluded that PPI therapy significantly improved LPR symptoms although they did not identify any difference in the post-treatment laryngoscopic findings [19].

Additionally, no significant events were observed in either of the two Cochrane reviews on the effect of PPIs on individual throat symptoms—cough [20] and dysphonia [21]. A meta-analysis suggests that PPIs have moderate superiority over placebo and the importance of diet as additional treatment. However, the considerable heterogeneity observed between the studies limits the formulation of a clear conclusion in many RCTs conducted using placebos [22].

A need is emerging from the literature to characterize patients accurately, since only a minority respond to antisecretory therapy, usually those with typical symptoms associated with extraesophageal GERD. With this aim, a recent multicenter study evaluated patients with chronic laryngeal symptoms and identified distinct phenotypes of patients, which included LPR/GERD with a hiatal hernia, LPR with mild GERD, no LPR or GERD, reflex cough, and mixed/possible obstructive esophagogastric junction. The authors demonstrated that individuals with LPR/GERD with hiatal hernia would likely be most responsive to PPI, followed by LPR and LPR with mild GERD and reflux cough [23]. This study confirmed that separating patients into distinct phenotypic categories may help to provide the most effective treatment for each patient.

Moreover, non-response to PPIs does not make it possible to exclude gastroesophageal reflux (GER) as a causative agent, especially considering the possibility of weakly acidic and biliary refluxes, which can be demonstrated by means of esophageal 24-h pH-impedance monitoring. In light of the increasing number of studies that identify acid and nonacid reflux as important causes of extraesophageal symptoms, our therapeutic approach has to evolve. The use of PPIs alone can be called into question, since these agents are less effective on nonacid or mixed reflux. Given the alkaline pH required for trypsin activity, the administration of high doses of PPIs may be associated with a worsening of complaints [24].

In this regard, in recent years, few studies have evaluated the efficacy of alginate therapy in patients with extraesophageal reflux symptoms. Alginates are an oral pharmacologic therapy that creates a barrier at the esophagogastric junction and a mechanical raft above the gastric contents to prevent gastroesophageal reflux events, whether acidic or nonacidic. In addition, alginates inhibit pepsin and bile salts. For this reason, alginates are particularly well-suited in the case of nonacid or mixed reflux or for patients with postprandial symptoms and, according to a recent article, could have similar efficiency to that of PPIs + alginate [25,26]. However, published RCT data are currently lacking.

To date, few published studies have shown the effects of products containing CS and HA in providing relief from typical GERD symptoms. Two small prospective placebo-controlled studies have shown that short-term treatment achieved significant and rapid symptom relief in patients with both erosive reflux disease [27] and NERD [28]. More recently, a prospective double-blind placebo-controlled trial conducted in several Italian centers showed that the combination of a PPI + HA-CS in syrup was able to relieve symptoms and improve the quality of life to a greater extent than PPI monotherapy [29]. A recent open-label, uncontrolled study showed that administration of orodispersible tablets containing CS, aluminum hydroxide, and HA improves typical and atypical clinical symptoms of non-erosive GERD and gastric juice-related biochemical parameters (e.g., neutrophils, lymphocytes, eosinophils, parietal cells, red blood cells, and exudate protein count) [30]. Moreover, in 2020, the conclusions of an exploratory study reported the first evidence that Gerdoff^®^ could effectively treat GER symptoms in patients not responding to PPI or alginate-based formulation [31].

### 4.2. Study Limitations

In this study, patients with extraesophageal symptoms and a recent diagnosis of gastroesophageal reflux disease were enrolled. The diagnosis of reflux disease with typical and atypical symptoms was based on the symptoms reported by the patients. No objective assessment of the association between reflux and extraesophageal symptoms was performed. Our study may be limited by the fact that no objective assessment of the association between reflux and extraesophageal symptoms was made and that diagnosis was based only on clinical presentations.

In clinical practice, the role of a gastroenterologist vis-à-vis patients referred for evaluation of suspected extraesophageal symptoms is to assess the possible association between reflux and symptoms. Non-GI investigations for ENT, pulmonary, and/or allergic conditions are essential, and in many cases, they should be the first diagnostic procedures performed, as extraesophageal symptoms often have a multifactorial or non-esophageal etiology. Combined pH and impedance monitoring is considered the best tool for characterizing gastroesophageal reflux because it is able to detect all types of reflux events (acidic, weakly acidic, and weakly alkaline) and determine the proximal extent of the refluxate within the esophagus. In our study, patients were not evaluated with 24-h pH monitoring, high-resolution manometry, and upper-GI endoscopy to assess the cause of the extraesophageal symptoms. Another possible limitation of the study is the great heterogeneity of the enrolled population. As a matter of fact, only some of the patients with extraesophageal symptoms enrolled in the study also showed typical reflux symptoms (such as regurgitation, reflux, and belching). An emerging fact in the literature is that patients with typical and atypical reflux symptoms respond better to PPI therapy than patients with extraesophageal symptoms alone [17]. It may be useful to bear this in mind when interpreting the results of our study. In this study, we did not include the assessment of examination findings as an outcome measure. Whilst this omission could be construed as a limitation of the study, as alluded to above, we would argue that in the LPR treatment setting, patients’ perceptions of symptoms are of greater clinical relevance than the endoscopic appearance of the larynx. Indeed, several studies have shown that laryngopharyngeal mucosal signs of reflux correlate poorly with patient-reported throat symptoms [32].

## 5. Conclusions

Extraesophageal reflux symptoms are increasingly common in the Western population and their clinical management is still controversial. Although therapy with PPIs represents the gold standard, to date, many patients are refractory to this treatment. It is therefore necessary to identify new drugs that can be used in the treatment of this clinical condition. This is the first published study on the efficacy of a medical device containing hyaluronic acid and chondroitin sulphate with antacid in the treatment of extraesophageal reflux symptoms and it shows that, in patients with extraesophageal symptoms and a recent diagnosis of GERD, treatment with Gerdoff^®^ + omeprazole leads to a greater, although not significantly so, reduction in symptoms than PPI treatment alone. The results of our study therefore suggest that medical devices containing combinations of hyaluronic acid and chondroitin sulphate may play an important role in the treatment of GERD with extraesophageal symptoms. We hope that our results will provide a starting point for—and stimulate—more experimentally robust RCTs going forward.

## Figures and Tables

**Figure 1 jcm-11-01890-f001:**
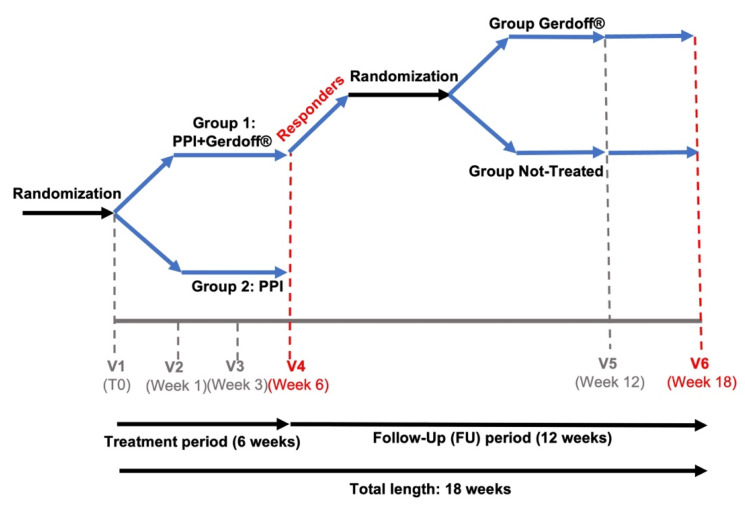
Study diagram. At baseline visit (visit V1), the enrolled patients were randomized with a 1:1 ratio in one of treatment groups: Group 1 (PPI + Gerdoff^®^) or Group 2 (PPI; control group). After the treatment period (6 weeks), the responders of Group 1 were further randomized with a 1:1 ratio in one of following groups defined according to the treatment: Group Gerdoff^®^ or Group Not-Treated. At week 12 from visit V1 (visit V5), the treatment was interrupted. The assessment of treatment effect maintenance was prolonged for a 6-week follow-up period (visit V6: week 18 from visit V1); PPI, proton pomp inhibitors.

**Figure 2 jcm-11-01890-f002:**
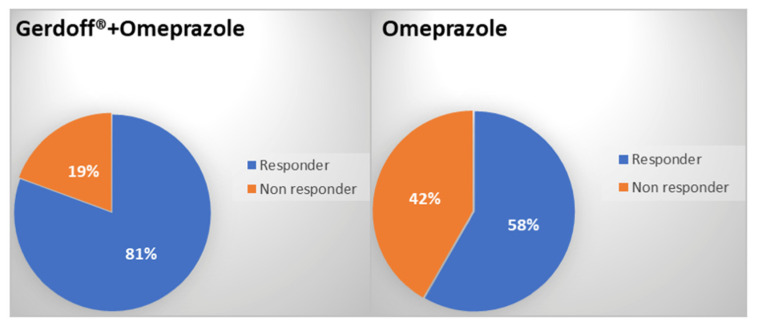
Distribution of responders at week 6 in the two groups. The percentages of responders at Week 6 in the treatment groups (Gerdoff^®^ + omeprazole or omeprazole) were reported (*p* = 0.0496 in the Chi-squared test).

**Table 1 jcm-11-01890-t001:** Summary of demographic and other baseline characteristics in the randomized population.

	Gerdoff^®^ + Omeprazole N = 35	Omeprazole N = 36	Total N = 71
Age (years)	*n* = 35	*n* = 36	*n* = 71
Mean (SD)	49.2 (15.22)	46.6 (14.36)	47.9 (14.74)
Gender, N (%)	*n* = 35	*n* = 36	*n* = 71
Male	13 (37.1%)	4 (11.1%)	17 (23.9%)
Female	22 (62.89)	32 (88.9%)	54 (76.1%)
Race, N (%)	*n* = 35	*n* = 36	*n* = 71
Caucasian	35 (100.0%)	36 (100.0%)	71 (100.0%)
Height (cm)	*n* = 29	*n* = 31	*n* = 60
Mean (SD)	167.5 (8.63)	165.0 (7.77)	166.2 (8.23)
Weight (kg)	*n =* 29	*n* = 31	*n* = 60
Mean (SD)	67.9 (15.65)	65.0 (13.90)	66.4 (14.72)
BMI (kg/m^2^)	*n* = 29	*n* = 31	*n* = 60
Mean (SD)	24.1 (4.80)	23.9 (4.96)	24.0 (4.84)
SBP (mmHg)	*n* = 31	*n* = 31	*n* = 62
Mean (SD)	119.6 (13.35)	123.3 (15.43)	121.5 (14.43)
DBP (mmHg)	*n* = 31	*n* = 31	*n* = 62
Mean (SD)	73.9 (8.68)	75.5 (11.28)	74.7 (10.06)
Heart rate (bpm)	*n* = 31	*n* = 31	*n* = 62
Mean (SD)	76.1 (13.81)	77.6 (10.02)	76.9 (11.99)
Respiratory rate (breaths/min)	*n* = 30	*n* = 30	*n* = 60
Mean (SD)	15.6 (1.92)	15.2 (2.31)	15.4 (2.12)

N = number of patients; *n* = number of observations; BMI, Body Mass Index; SBP, systolic blood pressure; DBP, diastolic blood pressure.

**Table 2 jcm-11-01890-t002:** Summary of total RSI score results.

Treatment Group	Study Visit	*n*	Mean	SD	Median	Min	Max
Gerdoff^®^ + omeprazole N = 35	Baseline/Screening (V1)	35	24.2	5.18	22	20	41
	1 week ± 1 day after baseline (V2)	32	15.3	7.86	15.5	2	32
	3 weeks ± 2 days after baseline (V3)	28	12.9	7.77	10.5	0	32
	6 weeks ± 2 days after baseline (V4)	31	7.9	6.04	7	0	21
Omeprazole N = 36	Baseline/Screening (V1)	36	26	4.96	24	20	37
	1 week ± 1 day after baseline (V2)	35	19.6	7.2	19	3	37
	3 weeks ± 2 days after baseline (V3)	35	15.3	8.08	16	2	32
	6 weeks ± 2 days after baseline (V4)	36	12.3	8.98	10.5	0	39

N = number of patients; *n* = number of observations; RSI, Reflux Symptoms Index.

**Table 3 jcm-11-01890-t003:** Summary of results for the primary performance endpoint (change in total RSI score between baseline and week 6).

Treatment Group	*n*	Mean	SD	SE of Mean	Median	Min	Max
Total	67	−14.8	9.28	1.13	−16.0	−38	13
Gerdoff^®^ + omeprazole	31	−16.2	7.45	1.34	−17.0	−38	0
Omeprazole	36	−13.7	10.58	1.76	−13.5	−34	13
Difference Gerdoff^®^ + omeprazole -omeprazole		−2.5	9.27	2.27	95% CI of the difference: −7.03 to 2.04		
Unpaired *t*-test: *p* = 0.2760 (*p* = 0.2679 with Cochran correction)							
Homogeneity of variance: *p* = 0.0536							

*n* = number of observations.

**Table 4 jcm-11-01890-t004:** Total RSI score results during the follow-up period.

Treatment	Visit	*n*	Mean	SD	Median	Min	Max
Gerdoff^®^	After 6 weeks (V4)	9	5.7	3	5	2	10
No treatment		8	5.5	4.07	6.5	0	10
Gerdoff^®^	After 18 weeks (V6)	9	4.2	3.53	4	0	11
No treatment		8	5.9	8.01	3	0	24

*n* = number of observations.

**Table 5 jcm-11-01890-t005:** Summary of changes in the total RSI score from baseline to any intermediate time point during the 6-week treatment period.

Variable	Treatment	*n*	Mean	SD	Median	Min	Max
Change at 1 week ± 1 day (V2)	Total	67	−7.6	7.21	−8.0	−32	6
	Gerdoff^®^ + omeprazole	32	−8.9	6.83	−10.0	−20	4
	Omeprazole	35	−6.4	7.42	−5.0	−32	6
Change at 3 weeks ± 2 days (V3)	Total	63	−11.1	7.86	−10.0	−30	9
	Gerdoff^®^ + omeprazole	28	−11.8	8	−13.0	−23	9
	Omeprazole	35	−10.7	7.82	−9.0	−30	1

*n* = number of observations.

**Table 6 jcm-11-01890-t006:** Summary of results regarding the changes in individual RSI item scores between baseline and any time point during the 6-week treatment period.

Item	Treatment	Visit	*n*	Mean	SD	Median	Min	Max
Hoarseness or voice problem	Gerdoff^®^ + omeprazole	Visit 1 (T0)	35	3.2	1.35	3	0	5
		Visit 2 (T1)	32	1.8	1.63	1	0	5
		Visit 3 (T3)	28	1.5	1.45	1	0	5
		Visit 4 (T6)	31	1.1	1.33	1	0	4
	Omeprazole	Visit 1 (T0)	36	2.8	1.54	3	0	5
		Visit 2 (T1)	35	2.4	1.63	2	0	5
		Visit 3 (T3)	35	1.8	1.57	2	0	5
		Visit 4 (T6)	36	1.6	1.54	1	0	5
Clearing the throat	Gerdoff^®^ + omeprazole	Visit 1 (T0)	35	3.4	1.22	4	1	5
		Visit 2 (T1)	32	2.5	1.5	2.5	0	5
		Visit 3 (T3)	28	2	1.4	2	0	5
		Visit 4 (T6)	31	1.4	1.39	1	0	5
	Omeprazole	Visit 1 (T0)	36	3.6	1	4	1	5
		Visit 2 (T1)	35	3.1	1.14	3	1	5
		Visit 3 (T3)	35	2.2	1.32	2	0	5
		Visit 4 (T6)	36	1.8	1.37	1	0	5
Excess throat mucus or post-nasal drip	Gerdoff^®^ + omeprazole	Visit 1 (T0)	35	2.7	1.76	3	0	5
		Visit 2 (T1)	32	2.3	1.58	3	0	5
		Visit 3 (T3)	28	2.1	1.53	2	0	5
		Visit 4 (T6)	31	1.4	1.31	1	0	5
	Omeprazole	Visit 1 (T0)	36	2.8	1.77	3	0	5
		Visit 2 (T1)	35	2.6	1.65	3	0	5
		Visit 3 (T3)	35	2.3	1.64	2	0	5
		Visit 4 (T6)	36	1.9	1.61	2	0	5

*n* = number of observations.

**Table 7 jcm-11-01890-t007:** Summary of total Likert scale scores at any time point.

Treatment Group	Study Visit	*n*	Mean	SD	Median	Min	Max
Gerdoff^®^ + omeprazole N = 35	Baseline/Screening (V1)	35	19	4.301	19	12	32
	1 week ± 1 day after baseline (V2)	32	13.9	5.975	15	3	29
	3 weeks ± 2 days after baseline (V3)	28	11.1	5.993	13	1	29
	6 weeks ± 2 days after baseline (V4)	31	7.4	5.667	8	0	31
	12 weeks ± 3 days after baseline (V5)	17	7.4	5.744	7	0	17
	18 weeks ± 3 days after baseline (V6)	17	4.2	5.238	2	0	17
Omeprazole N = 36	Baseline/Screening (V1)	36	21	4.557	18	12	32
	1 week ± 1 day after baseline (V2)	35	16.4	5.977	13.5	3	29
	3 weeks ± 2 days after baseline (V3)	35	14.4	6.735	12	1	25
	6 weeks ± 2 days after baseline (V4)	36	11.6	8.083	6	0	18

N = number of patients; *n* = number of observations.

**Table 8 jcm-11-01890-t008:** Summary of adverse events (AEs) in the two groups.

	Gerdoff^®^ + Omeprazole N = 35	Omeprazole N = 36
No. of AEs	28	29
No. (%) of patients with AEs	14 (40.0%)	12 (33.3%)
No. of SAEs	0	1
No. (%) of patients with SAEs	0 (0.0%)	1 (2.8%)
Intensity of AEs: No. (%) of AEs		
Mild	23 (82.1%)	23 (79.3%)
Moderate	5 (17.9%)	6 (20.7%)
Action taken: No. (%) of AEs		
None	22 (78.6%)	18 (50.0%)
Drug therapy	6 (17.1%)	8 (22.2%)
Non-drug therapy	0 (0.0%)	1 (2.8%)
Temporary interruption or dose adjustment	0 (0.0%)	1 (2.8%)
Hospitalization	0 (0.0%)	1 (2.8%)
Outcome: No. (%) of AEs		
Resolved	25 (89.3%)	23 (79.3%)
Unresolved	0 (0.0%)	2 (5.6%)
Unknown	3 (10.7%)	4 (11.1%)

N = number of patients.

## Data Availability

The data supporting reported results are available from the corresponding author on reasonable request.

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
