# Peer review of "The Effect of Hyaluronic Acid and Chondroitin Sulphate-Based Medical Device Combined with Acid Suppression in the Treatment of Atypical Symptoms in Gastroesophageal Reflux Disease"

_jcm, 2022, doi:10.3390/jcm11071890_

Round 1
Reviewer 1 Report
The paper "The effect of Gerdoff® combined with acid suppression in the treatment of Atypical Symptoms in Gastroesophageal Reflux Disease" is well elaborated, with clear objectives and very sound statistics.
However, the references are old more than 5 years, and even that the Montreal classification of GERD is mentioned, "Vankil N, Veldhuyzen van Zanten S, Kahrilas P, et al. The Montreal definition and classification of gastroesophageal reflux disease (GERD)- a global evidence based consensus. Am J Gastroenterol. 2006;101:1900–20" it is not included in the references. I would suggest to revise the references. It is correct, most of the guidelines are old now, but there are some new studies.
Also I would comment more on the fact that the diagnosis of GERD was based only on symptoms.
Reviewer 2 Report
Main Comment:
(1) This manuscript deals with the effect of a hyaluronic acid- and chondroitin sulphate-based product combined with acid suppression in the treatment of atypical symptoms in gastroesophageal reflux disease. The presented study has several limitations (see end of the Discussion); it can just be seen as a "starting point" for more robust trials.
Specific Comments/Suggestions:
(2) There is a problem with Table 1: "Height (cm)" – one line is missing here; "Age (years): Mean (SD): 167.5 (8.63), 165.0 (7.77), 166.2 (8.23)" – this is obviously erroneous.
(3) Line 2 (title): I would suggest not using the brand name in the title. Besides, capitalization is used inconsistently.
(4) Line 116: Study design: Which randomization technique was used?
(5) Line 164: change in "upper symptoms"?
(6) Lines 191-193: "Thirteen (37.1%) patients in the Gerdoff+omeprazole group and 2 (5.6%) in the omeprazole group dropped out of the study during the 6-week treatment period due to withdrawal of consent" – why was the drop-out rate so high in the study group?
(7) Line 200: "with reversed rates for females (62.8% and 88.9% of patient, p = 0.0102)": "patient" -> patients; however, this passage can be omitted completely.
(8) Line 253: "did not resulted" -> did not result.
(9) Figure legend 2: "The number and percentage of responders at Week 6 in the treatment groups (Gerdoff+omeprazole or omeprazole) were reported" – the figure contains just the percentages (not the numbers).
(10) Lines 289-290: "The majority of patients in both groups gave an excellent or a good satisfaction judgment for treatment at any post-baseline timepoint" – please provide the percentages.
(11) Line 358: "upper GERD symptom"?
(12) Line 384/385: "only a few are randomized controlled trial (RCTs)" -> only a few are randomized controlled trials (RCTs).
(13) Line 473 and Table S1: "no statistical different items" -> no statistically different items.
(14) There is also a problem with Table S1 (frameshift).
(15) Footer of Table S1: "N = number of patients" – no "N" is contained in this table.
(16) Citations: Line 74; line 412: "[8], [9]" -> [8,9]; "[21], [22]" -> [21,22].
